# Impact of assessment and intervention by a health and social care professional team in the emergency department on the quality, safety, and clinical effectiveness of care for older adults: A randomised controlled trial

Marica Cassarino[1,2], Katie Robinson[1], Dominic Trépel[3], Íde O'Shaughnessy[4], Eimear Smalle[4], Stephen White[4], Collette Devlin[1], Rosie Quinn[5], Fiona Boland[6], Marie E. Ward[7], Rosa McNamara[8], Fiona Steed[9], Margaret O'Connor[10,11], Andrew O'Regan[11], Gerard McCarthy[12], Damien Ryan[13], Rose Galvin[1]*

1 School of Allied Health, Faculty of Education and Health Sciences, Health Research Institute, Ageing Research Centre, University of Limerick, Castletroy, Ireland, 2 School of Applied Psychology, University College Cork, Cork, Ireland, 3 Trinity Institute of Neurosciences, School of Medicine, Trinity College Dublin, Dublin, Ireland, 4 Emergency Department, University Hospital Limerick, Dooradoyle, Limerick, Ireland, 5 Emergency Department, Our Lady of Lourdes Hospital Drogheda, Drogheda, Ireland, 6 Data Science Centre, Royal College of Surgeons in Ireland, Dublin, Ireland, 7 School of Psychology, Trinity College, the University of Dublin, Dublin, Ireland, 8 Emergency Department, St. Vincent University Hospital, Dublin, Ireland, 9 Department of Physiotherapy, University Hospital Limerick, Dooradoyle, Limerick, Ireland, 10 Department of Ageing and Therapeutics, University Hospital Limerick, Dooradoyle, Limerick, Ireland, 11 School of Medicine, Faculty of Education and Health Sciences, University of Limerick, Castletroy, Ireland, 12 Emergency Department, Cork University Hospital, Cork, Ireland, 13 Limerick EM Education Research Training (ALERT), Emergency Department, University Hospital Limerick, Dooradoyle, Limerick, Ireland

* rose.galvin@ul.ie

## Abstract

### Background

Older adults frequently attend the emergency department (ED) and experience high rates of adverse events following ED presentation. This randomised controlled trial evaluated the impact of early assessment and intervention by a dedicated team of health and social care professionals (HSCPs) in the ED on the quality, safety, and clinical effectiveness of care of older adults in the ED.

### Methods and findings

This single-site randomised controlled trial included a sample of 353 patients aged ≥65 years (mean age = 79.6, SD = 7.01; 59.2% female) who presented with lower urgency complaints to the ED a university hospital in the Mid-West region of Ireland, during HSCP operational hours. The intervention consisted of early assessment and intervention carried out by a HSCP team comprising a senior medical social worker, senior occupational therapist, and senior physiotherapist. The primary outcome was ED length of stay. Secondary outcomes included rates of hospital admissions from the ED; hospital length of stay for admitted patients; patient satisfaction with index visit; ED revisits, mortality, nursing home admission,

**Data Availability Statement:** The data used in this study are available on the Open Science Framework at the public link https://osf.io/5wpjm/.

**Funding:** This research is supported by the Health Research Board of Ireland through the Research Collaborative for Quality and Patient Safety (RCQPS) 2017, granted to the Principal Investigator (RG). The sponsor was not involved in the design of the study and collection, analysis, interpretation of data, or in writing the manuscript.

**Competing interests:** The authors have declared that no competing interests exist.

**Abbreviations:** ADLs, activities of daily living; ANCOVA, analysis of covariance; CI, confidence interval; CONSORT, Consolidated Standards of Reporting Trials; ED, emergency department; EM, emergency medicine; EQ-5D-5L, EuroQoL 5-dimension 5-level; GP, general practitioner; HSCP, health and social care professional; HSE, Health Service Executive; ISAR, Identification of Seniors at Risk; MNA, Mini Nutritional Assessment; MRC, Medical Research Council; MTS, Manchester Triage System score; OR, odds ratio; PET, Patient Experience Time; PSQ-18, 18-item Patient Satisfaction Questionnaire.

and unscheduled hospital admission at 30-day and 6-month follow-up; and patient functional status and quality of life (at index visit and follow-up). Demographic information included the patient's gender, age, marital status, residential status, mode of arrival to the ED, source of referral, index complaint, triage category, falls, and hospitalisation history. Participants in the intervention group ($n = 176$) experienced a significantly shorter ED stay than the control group ($n = 177$) (6.4 versus 12.1 median hours, $p < 0.001$). Other significant differences (intervention versus control) included lower rates of hospital admissions from the ED (19.3% versus 55.9%, $p < 0.001$), higher levels of satisfaction with the ED visit ($p = 0.008$), better function at 30-day ($p = 0.01$) and 6-month follow-up ($p = 0.03$), better mobility ($p = 0.02$ at 30 days), and better self-care ($p = 0.03$ at 30 days; $p = 0.009$ at 6 months). No differences at follow-up were observed in terms of ED re-presentation or hospital admission. Study limitations include the inability to blind patients or ED staff to allocation due to the nature of the intervention, and a focus on early assessment and intervention in the ED rather than care integration following discharge.

## Conclusions

Early assessment and intervention by a dedicated ED-based HSCP team reduced ED length of stay and the risk of hospital admissions among older adults, as well as improving patient satisfaction. Our findings support the effectiveness of an interdisciplinary model of care for key ED outcomes.

## Trial registration

ClinicalTrials.gov NCT03739515; registered on 12 November 2018.

---

## Author summary

### Why was the study done?

- Some studies suggest that older patients presenting to the emergency department (ED) could benefit from receiving early assessment and intervention by a dedicated health and social care professional (HSCP) team, particularly in terms of safer discharges and increased patient and staff satisfaction.

- To date, no methodologically robust studies exist that have tested the effectiveness of this interdisciplinary model of ED care for older adults.

- In this randomised controlled trial, we evaluated the impact of an ED-based HSCP team dedicated to older patients on ED length of stay, incidence of hospital admissions, and other measures of quality and safety of care.

### What did the researchers do and find?

- Compared to those receiving usual ED care, patients who were treated by the HSCP team spent less time in the ED, had lower rates of hospital admission, and were satisfied with their care, as well as reporting better function at follow-up.

- We observed no differences in the numbers of patients re-presenting to the ED at 30 days or 6 months.

### What do these findings mean?

- Having a dedicated HSCP team that provide early assessment and intervention to older people with lower urgency conditions can improve quality and timeliness of ED care.

- Further research is needed to understand whether these benefits can extend to other patient populations, as well as clarifying implications for integration of care post-discharge from the ED.

## Introduction

Internationally, emergency departments (EDs) face significant challenges in delivering high-quality and timely patient care set against a background of increasing patient numbers and limited hospital resources [1,2]. A growing ageing population and a higher number of individuals with multimorbidity are among the main demographic drivers of increases in ED attendances [3,4], which, in turn, lead to ED crowding. Research has demonstrated that ED crowding contributes to a reduction in the quality of patient care, delays in commencement of treatment, increased mortality, poorer adherence to recognised clinical guidelines, and ultimately poorer health outcomes [4,5].

The implementation of quality improvement strategies that can help to streamline patient flow in the ED has been shown enhance quality of care [6]; these strategies are especially important for older adults, whose health outcomes are negatively impacted by frequent ED attendances [7,8] and ED crowding. Increasingly, evidence from international studies has supported the introduction of teams of health and social care professionals (HSCPs) to the ED to provide specialised interdisciplinary care, which can contribute to more timely and safer decision-making [9–15]. A recent systematic review [16] indicated that care coordination teams of HSCPs (including physiotherapists, occupational therapists, and medical social workers), which provide early assessment and intervention to older adults in the ED can lead to safer discharges and increase patient and staff satisfaction; however, the quality of the evidence is mixed, primarily due to weaknesses in study designs and heterogeneity of patient groups and outcomes of interest. On the other hand, ED stakeholders have reported positive perceptions of HSCPs working in the ED, in particular the added value of having HSCPs with specialised skills working in an interdisciplinary manner to provide timely care to older adults, while at the same time reducing the workload of the ED medical staff [17,18]. This evidence suggests that ED-based HSCP teams could be a viable and effective solution to improve the flow and outcomes of ED patients, particularly those who might benefit the most from interdisciplinary holistic assessment [19]. However, no robust investigations of the effectiveness of this model of care have been carried out to date [16].

Building on the Medical Research Council (MRC) framework for the development and evaluation of complex interventions [20], this randomised controlled trial [21] aimed to evaluate the impact of a dedicated team of HSCPs in the ED on the quality, safety clinical, and cost-effectiveness of care of older adults in the ED. Specifically, this study describes the findings on

the clinical effectiveness of the intervention in terms of key ED and patient outcomes. An analysis of cost-effectiveness of the intervention is presented separately [22].

## Methods/design

### Design and setting

The study represents a single-centre parallel group randomised controlled trial, which investigates the clinical effectiveness of early assessment and intervention by a dedicated HSCP team on older adults in the ED of a university teaching hospital with a large catchment area in the Mid-West of the Republic of Ireland. The university teaching hospital sits within a larger hospital group with 6 hospital sites but all function as a single hospital system caring for a substantially rural population of approximately 385,000. The university teaching hospital is the only hospital in the group that has a full 24/7/365 emergency care and critical care service and has 455 inpatient beds.

This report adheres to the Consolidated Standards of Reporting Trials (CONSORT) guidelines [23]; the CONSORT Checklist is included in the S1 CONSORT Checklist and the Participant Flowchart is presented in Fig 1. The trial protocol is registered on ClinicalTrials.gov (NCT03739515) and is published elsewhere [21].

The study received ethics approval from the Health Service Executive (HSE) Mid-Western Regional Hospital Research Ethics Committee (Ref: 103/18).

### Participants and recruitment

Patients were eligible for participation in the study if they were aged ≥65 years, medically stable, able to provide written consent, and presented to the ED during HSCP operational hours with one of the index complaints included in Table 1. Eligible patients were identified by the dedicated HSCP team employed in the project through the ED triage system and via consultation with the ED medical staff. Prospective patients who had a Manchester Triage System score (MTS) of 3 to 5 (respectively, urgent, standard, or nonurgent) and who were categorised at triage with limb problems, falls, unwell adult, back pain, urinary problems, or ear/facial problems were invited to take part. The HSCP team completed baseline measurement and randomisation prior to assessment by a doctor in emergency medicine (EM). Patients who presented with an MTS score of 2 (very urgent) or any of the MTS categories not listed above were invited to take part following consultation with the EM doctor and completion of diagnostic work-up, to determine suitability and indication for HSCP assessment, e.g., MTS score 2 presenting with falls, life-threatening condition/injury out-ruled and medically stable. All potential participants were provided with an information sheet, briefed on the study, and given time to decide or consult with a caregiver (if present). Patients who were willing to take part were then asked to read and sign a written informed consent form. Consent procedures and mechanisms relating to data controlling and processing complied with the EU General Data Protection Regulation 2016/679 and with the Data Protection Act 2018 [(Section 36(2)) (Health Research) Regulations 2018]. The study received ethical approval from the HSE Mid-Western Regional Hospital Research Ethics Committee (Ref: 103/18).

### Baseline data collection

After consenting to take part, each participant underwent a baseline assessment with a member of the HSCP team; this had an approximate duration of 30 minutes. Demographic information included the patient's gender, age, marital status (single, married, widowed,

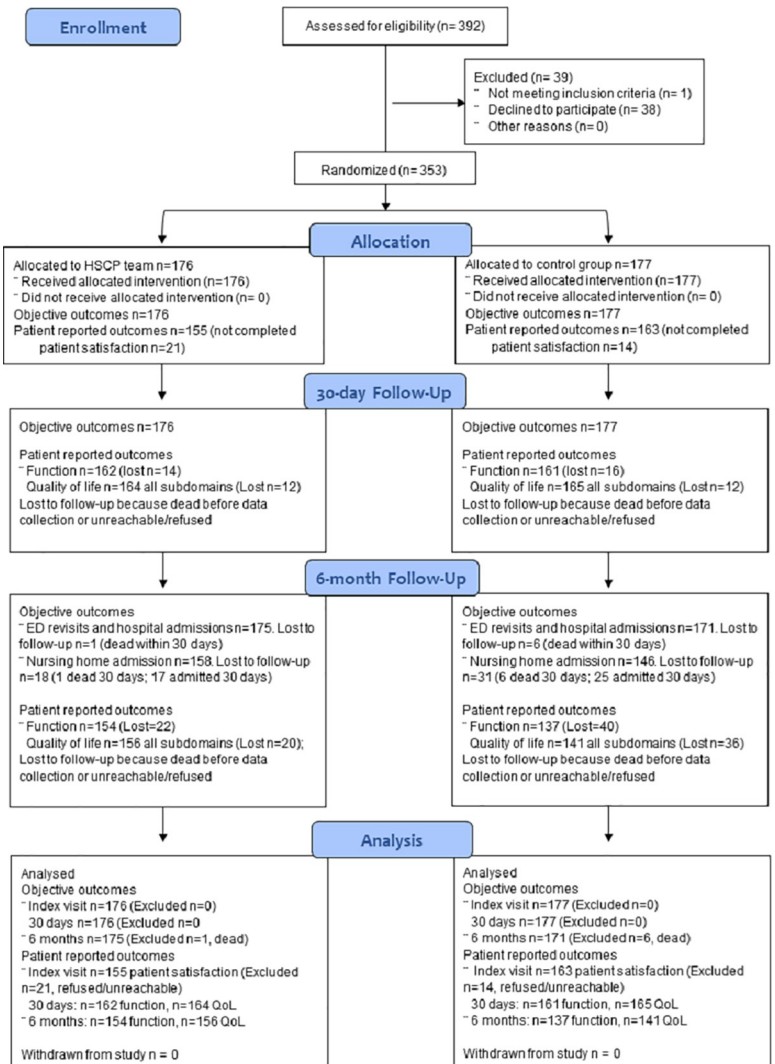

**Fig 1. Participant flow chart, based on the CONSORT 2010 diagram (23).** ED, emergency department; HSCP, health and social care professionals; QoL, quality of life.

separated), residential status (living alone versus living with others), mode of arrival to the ED (private transport, ambulance, public transport, other), source of referral (self-referral, general practitioner, injury unit, walk-in clinic, other), index complaint (as per Table 1), and triage category. The health assessment comprised of the following measurements: functional status measured through the Barthel index for activities of daily living (ADLs) [25]; health-related quality of life through the EuroQoL survey 5-dimension and 5-levels form (EQ-5D-5L) [26]; risk of adverse outcomes measured via the Identification of Seniors At Risk (ISAR) screening tool [27]; frailty via the Rockwood's Clinical Frailty Scale [28]; nutritional status through the Mini Nutritional Assessment (MNA) short form [29]; whether the person had experienced any falls in the 3 months prior to the ED visit; whether the person had been hospitalised in the 6 months prior to the ED visit. Functional status and quality of life were also assessed at follow-up.

**Table 1. Patient eligibility criteria.**

| Inclusion criteria | Exclusion criteria |
|---|---|
| Aged ≥65 years | Aged under 65 years |
| MTS 3–5 | MTS 1–2* |
| Off baseline mobility and functional status | Neither the patient nor the carer can communicate in English sufficiently to complete informed consent or baseline assessment |
| Capacity and willingness to provide informed consent | Lacking capacity to provide informed consent** |
| Presenting during HSCP operational hours (8 AM–5 PM Monday–Friday) | Presenting outside HSCP operational hours (5 PM–8 AM or on Saturday/Sunday) |
| Presenting with any of the following complaints, as per MTS [24]: *Before medical work-up*: Limb problems Falls Unwell adult Back pain Urinary problems Ear and facial problems | Presenting with complaints other than described in the inclusion list. |

EM, emergency medicine; HSCP, health and social care professional; MTS, Manchester Triage System.

*MTS score 1–2 only recruited after EM diagnostic work-up and suitability for HSCP assessment determined.

**In cases where there was a clinical concern regarding capacity to consent, the 4AT tool was used to screen for cognitive impairment and in participants where there was evidence of moderate-profound impairment, the patient's nominated contact person was contacted for consent.

## Randomisation

Computer-generated random numbers in blocks of 20 were created by a researcher independent of the recruitment process (MC) using an internet-based system (https://www.randomizer.org/) and placed in sealed opaque envelopes. These numbers were securely stored in the presealed envelopes in a locked drawer in the research facility on the hospital site. Allocation was revealed by a research nurse (CD) after recruitment of eligible participants and the conduct of the baseline assessment by accessing and opening the next envelope in the sequence and providing the randomisation information to the research team and patient simultaneously. After allocation was revealed, participants were assigned to either the experimental group (HSCP intervention) or the control group (routine care). Due to the nature of the study, blinding of participants and personnel was not possible after allocation to intervention or control.

## Experimental and control interventions

**Intervention.** Participants in the intervention group were assessed by one or more members of the dedicated HSCP team (one senior physiotherapist, one senior occupational therapist, and one senior medical social worker). The assessment included, but was not be limited to, an interdisciplinary assessment of functional and mobility status, cognition, and psychosocial needs. Members of the HSCP team were guided by their clinical expertise, scope, and codes of professional practice. Details of the common domains covered by the HSCP team in their assessment and intervention are presented in S1 Intervention Characteristics. Interventions prescribed by the HSCP team were based on subjective and objective assessment of patients and included prescription of mobility aids and enabling ADL equipment, provision of home exercise programmes, education of self-management strategies, and onward referral to alternative care pathways, as deemed appropriate. All assessments and interventions were

included in the ED chart of individual participants. The HSCP team worked together with the ED medical and ED nursing team to develop an optimal recommendation for the patient in terms of discharge and relevant interventions. The HSCP team had a dedicated cubicle in the ED where they performed assessment and intervention, as appropriate. However, a number of assessments were completed trolley side after individual risk assessment.

**Control group.** The comparison group received routine care provided by the ED medical and nursing staff for the duration of their stay in the ED. Prior to this study, there was no dedicated team of HSCPs to assess and intervene with older adults who presented to the ED at this hospital. Ad hoc services were provided by HSCPs if they were requested by a member of the team in the ED. This process continued for the duration of the trial, but services to the control group were provided by HSCPs that were not part of the dedicated HSCP team in the ED.

## Outcomes

The primary outcome of the study was the duration of stay in the ED in hours from the time of arrival in the ED to the time of discharge. Secondary outcomes included the rate of hospital admission from the ED, hospital length of stay for admitted patients, and patient satisfaction with the ED index visit assessed through the 18-item Patient Satisfaction Questionnaire (PSQ-18) [30] at the time of the visit. Follow-up outcomes included the number of unplanned ED reattendances, unscheduled hospital visits, and nursing home admission within 30 days and 6 months of the initial index visit. Functional status [25] and quality of life [26] were also assessed at follow-up (30 days and 6 months).

## Data collection

The research nurse collected data on objective measures (e.g., ED length of stay, admission rates, ED revisits, unscheduled hospital admissions) using the hospital routine databases. A hospital attendance or admission at any of the 3 other local hospitals in this hospital group would also be captured on this database. Patient-reported outcomes were assessed via questionnaires either in person (at the time of the ED index visit) or via a telephone call at follow-up: These included patient satisfaction (index visit), functional status, mortality, and quality of life (30 days and 6 months). Each participant in the study was assigned a unique numerical code in order to link data collected at baseline to the follow-up. Data on healthcare utilisation were also collected at 30 days and 6 months and will be included in a separate cost-effectiveness publication (in preparation at the time of submission). No reminder of indications provided at discharge was given during the follow-up phone calls.

## Data analysis

Statistical analyses were conducted using Stata 12 MP (StataCorp LP, Texas). Prespecified analyses can be found in the trial protocol [21]. Participants' characteristics at baseline and follow-up stratified by intervention and control group are described using means and standard deviation for continuous measures (e.g., patient's age, quality of life subdomains); median and interquartile ranges for continuous measures, which showed evidence of skewness (e.g., ED length of stay); and frequencies and percentages for categorical measures (e.g., hospital admissions, ED revisits at follow-up).

Differences between the 2 groups in terms of ED length of stay and hospital length of stay were analysed using the Wilcoxon–Mann–Whitney test. Logistic regression was used to explore differences between the intervention and control in hospital admissions at the index visit, as well as 30-day and 6-month ED revisits, unscheduled hospital, and nursing home admissions. The models were adjusted for patient's age and ISAR score at baseline and results

were presented as odds ratios (ORs) and 95% confidence intervals (CIs). Patients' function and each of the quality of life subdomains at follow-up were explored using an analysis of covariance (ANCOVA) fitting a linear regression model adjusting for patient's age, ISAR score, baseline Barthel ADL index (for function), and baseline quality of life (for quality of life). The differences in patient satisfaction scores with their index visit were analysed using an independent samples *t* test with 95% CI.

## Sample size

The sample size was calculated based on our primary outcome (ED length of stay) using G*Power version 3.1. Drawing on the data from the Patient Experience Time (PET) database employed in the hospital ED, the average ED length of stay for patients aged 65 and older for the period 2016 to 2017 was 13.95 hours (standard deviation = 12.49 hours) [21]. Estimating a 40% decrease in ED length of stay in the intervention group (mean = 8.37 hours), as observed in previous studies [15], and with a 20% attrition rate to follow-up, a sample size of 258 patients (129 in each group) was required to achieve 90% power with two-tailed tests at an alpha level of 0.05.

## Patient and public involvement

The research questions and outcome measures for this trial were informed by a participatory study where inputs from multiple stakeholders, including ED patients and caregivers, were gathered with regard to this model of care [18]; patients' insights were also useful to guide decisions on the dissemination of the findings of this intervention. Patients were involved in this trial as study participants at the time of data collection; given the nature of the study and the setting of data collection, it was felt that involving patients as coinvestigators may cause a burden for individuals who may already present vulnerabilities. Nonetheless, patients' feedback on the intervention was gathered as part of the satisfaction questionnaire.

## Results

Participants were recruited from December 2018 until the end of May 2019. As shown in the trial flow in Fig 1, a total of 392 patients were initially identified as eligible and 353 were enrolled in the trial: 176 in the intervention group (HSCP team) and 177 in the control group (usual ED care). Of the 39 patients not recruited, 38 refused to take part because they were not interested (65%) or did not feel well enough to participate (reasons cited included pain, tiredness, or distress) (35%); one patient was excluded presented as a return rather than a new visit.

At the index visit, data relating to ED length of stay and hospital admissions was collected for all 353 patients. A total of 35 patients (13 in the control and 21 in the intervention group) did not complete the satisfaction questionnaire.

At 30-day follow up, data on objective outcomes (ED revisits, unscheduled hospital admissions, and nursing home admissions) were available for all participants; data on function were available for 323 patients ($n = 30$ lost to follow-up because dead or refused/unreachable); and data on quality of life subdomains were available for 329 patients ($n = 24$ lost to follow-up). At 6 months, analyses for ED revisits and unscheduled hospital admissions included 346 patients (excluding 7 participants who had died within 30 days); analyses for nursing home admissions included 304 patients (excluding 7 participants who had died within 30 days and 42 already admitted to nursing homes within 30 days). In terms of patient-reported outcomes, data for function were available for 291 patients ($n = 62$ lost to follow-up because dead or refused/unreachable), and data for quality of life across all subdomains were present for 297 patients ($n = 56$ lost to follow-up).

Table 2 presents the participants' baseline characteristics by group and for the overall sample, including both demographic and health measures. The sample mean age was 79.6 years (SD = 7.01 years), with 59.2% female participants. Patients included in the trial presented an MTS score ranging from 2 "very urgent" to 4 "standard," while no patients with a score of 1 (immediate) or 5 (nonurgent) were part of the sample. In terms of baseline differences between the 2 groups (see Table 2), we observed that patients in the control group were slightly older than those in the intervention group and more likely to be in triage category 3, whereas participants in the intervention group reported more pain or discomfort than those in the control group.

Patients' outcomes related to their index visit are presented in Table 3. Participants in the intervention group spent almost 50% less time in the ED when compared to the control group (median hours 6.43 versus 12.10; Mann–Whitney Z = 4.82, $p < 0.001$) and had a significant reduced risk of hospital admissions (OR = 0.18, 95% CI 0.11 to 0.31, $p < 0.001$). To account for the potential influence of clinical severity, and considering potential baseline differences in triage category, age, and health status, we conducted sensitivity analyses controlling for triage category, age, ISAR score, nutritional status, and clinical frailty score, both for ED length of stay (using linear regression) and for rates of hospital admissions (logistic regression); these can be seen in S1 Sensitivity Analyses. For both measures, the differences between intervention and control group were statistically significant even when considering potential differences in clinical conditions. Participants in the intervention group reported higher satisfaction with the care received in the ED when compared to the control group ($p = 0.008$). There was no evidence of a difference between the 2 groups with regard to the length of stay in hospital for patients who were ultimately admitted ($p = 0.32$).

At 30-day follow-up (Table 4), after adjusting for covariates, there was no evidence of a difference between the 2 groups in terms of unscheduled ED revisits ($p = 0.23$), unscheduled hospital admissions ($p = 0.92$), or nursing home admissions ($p = 0.43$). Participants in the intervention group reported higher functional status than those in the control group ($p = 0.003$). Sensitivity analyses of the delta of Barthel scores (30 days minus index visit) indicated that participants in the intervention group had a significant improvement from baseline to 30 days (mean = 0.21, SD = 2.91) compared to those in the control group (mean = −0.82, SD = 3.14); diff = −1.02, t(321) = −3.04, $p = 0.003$, Cohen's d = 0.34. In terms of quality of life, patients in the intervention group reported better mobility ($p = 0.02$) and better self-care ($p = 0.03$) when controlling for age and ISAR score, but there was no evidence of significant differences in quality of life in terms of ability to perform usual activities ($p = 0.22$), discomfort ($p = 0.97$), or anxiety/depression ($p = 0.47$). Sensitivity analyses on the delta score for each of the EuroQoL subdimensions (30 days minus baseline) indicated no statistically significant differences between intervention and control group for any of the subdimensions.

A similar pattern of results was observed at 6 months (Table 5), although there was a significantly lower rate of unscheduled hospital admissions for patients in the intervention when compared to the control group ($p = 0.02$). Similar to 30-day follow-up, patients in the intervention group reported better function ($p = 0.04$). However, when we considered changes in Barthel scores between 30 days and 6 months, we noted that there were no significant differences between the 2 groups (intervention, mean = −0.03, SD = 2.32; control, mean = 0.08, SD = 2.58; t(282) = 0.39, $p = 0.69$). When considering quality of life, we found better self-care in the intervention group than in control group ($p = 0.03$) 6 months after their index visit, although the differences were small in size, but no differences in other measures of quality of life. When running sensitivity analyses on changes in quality of life between 30 days and 6 months, no significant differences were noted for any of the subdimensions.

**Table 2. Baseline characteristics.**

| Characteristic | Control (*n* = 177) | Intervention (*n* = 176) | Overall (*N* = 353) |
|---|---|---|---|
| Female, n (%) | 101 (57.1) | 108 (61.4) | 209 (59.2) |
| Age, mean ± SD | 80.6 ± 6.82 | 78.6 ± 7.08 | 79.6 ± 7.01 |
| Marital status, n (%) | | | |
| Single | 28 (15.8) | 23 (13.1) | 51 (14.5) |
| Married | 65 (36.7) | 70 (39.8) | 135 (38.2) |
| Divorced | 6 (3.4) | 5 (2.8) | 11 (3.1) |
| Widowed | 78 (44.1) | 75 (42.6) | 153 (43.3) |
| Unknown | 0 | 3 (1.7) | 3 (0.9) |
| Residential status, n (%) | | | |
| Lives alone | 79 (44.6) | 74 (42.1) | 153 (43.3) |
| Lives with family | 95 (53.7) | 88 (50.0) | 183 (51.8) |
| Nursing home resident | 1 (0.6) | 6 (3.4) | 7 (2.0) |
| Other* | 2 (1.1) | 8 (4.6) | 10 (2.8) |
| Mode of entry, n (%) | | | |
| Ambulance | 86 (48.6) | 85 (48.3) | 171 (48.4) |
| Private transport | 88 (49.7) | 89 (50.6) | 177 (50.1) |
| Public transport | 2 (1.1) | 2 (1.1) | 4 (1.1) |
| Walk-in | 1 (0.6) | 0 | 1 (0.3) |
| Referral, n (%) | | | |
| GP services | 71 (40.1) | 58 (33.0) | 129 (36.5) |
| Self-referral | 102 (57.6) | 115 (65.3) | 217 (61.5) |
| Nursing home | 3 (1.7) | 1 (0.6) | 4 (1.1) |
| Walk-in clinic | 1 (0.6) | 1 (0.6) | 2 (0.6) |
| Other** | 0 | 1 (0.6) | 1 (0.3) |
| Index complaint, n (%)*** | | | |
| Limb problems | 61 (34.5) | 73 (41.5) | 134 (38.0) |
| Unwell adult | 39 (22.0) | 24 (13.6) | 63 (17.9) |
| Falls | 28 (15.8) | 26 (14.8) | 54 (15.3) |
| Back pain | 15 (8.5) | 22 (12.5) | 37 (10.5) |
| Other† | 34 (19.2) | 31 (17.6) | 65 (18.4) |
| Triage category, n (%) | | | |
| 2 | 11 (6.2) | 25 (14.2) | 36 (10.2) |
| 3 | 155 (87.6) | 140 (79.6) | 295 (83.6) |
| 4 | 11 (6.2) | 11 (6.3) | 22 (6.2) |
| Falls in the past 3 months, n (%) | 92 (52.3) | 96 (54.6) | 188 (53.4) |
| Hospitalised in the past 6 months, n (%) | 63 (35.59) | 50 (28.57) | 113 (32.10) |
| Barthel index, mean ± SD | 16.3 ± 3.68 | 16.8 ± 3.13 | 16.6 ± 3.43 |
| EQ-5D-5L, mean ± SD | | | |
| Mobility (*n* = 353) | 2.86 ± 1.35 | 2.64 ± 1.29 | 2.75 ± 1.32 |
| Self-care (*n* = 353) | 2.54 ± 1.34 | 2.38 ± 1.33 | 2.45 ± 1.34 |
| Usual activities (*n* = 353) | 2.99 ± 1.27 | 2.80 ± 1.26 | 2.91 ± 1.27 |
| Pain/discomfort (*n* = 353) | 2.57 ± 1.17 | 2.84 ± 1.16 | 2.70 ± 1.17 |
| Anxiety/depression (*n* = 351) | 1.60 ± 0.89 | 1.53 ± 0.84 | 1.57 ± 0.86 |
| ISAR score, n (%) | | | |
| <2 | 34 (19.2) | 41 (23.3) | 75 (21.3) |
| ≥2 | 143 (80.8) | 135 (76.7) | 278 (78.7) |
| Clinical Frailty Score, n (%) | | | |

*(Continued)*

**Table 2.** (Continued)

| Characteristic | Control (n = 177) | Intervention (n = 176) | Overall (N = 353) |
|---|---|---|---|
| 1. Very fit | 4 (2.3) | 8 (4.5) | 12 (3.4) |
| 2. Well | 20 (11.3) | 15 (8.5) | 35 (9.9) |
| 3. Managing well | 34 (19.2) | 48 (27.3) | 82 (23.2) |
| 4. Vulnerable | 43 (24.3) | 51 (28.9) | 94 (26.6) |
| 5. Mildly frail | 36 (20.3) | 31 (17.6) | 67 (18.9) |
| 6. Moderately frail | 35 (19.7) | 19 (10.8) | 54 (15.3) |
| 7. Severely frail | 4 (2.3) | 3 (2.3) | 8 (2.3) |
| 9. Terminally ill | 1 (0.6) | 0 | 1 (0.3) |
| Nutritional status, n (%) | | | |
| Malnourished | 16 (9.0) | 11 (6.3) | 27 (7.7) |
| Risk of malnutrition | 58 (32.8) | 41 (23.3) | 99 (28.1) |
| Normal status | 103 (58.2) | 124 (70.5) | 227 (64.3) |

EQ-5D-5L, EuroQoL 5-dimension 5-level; IQR, interquartile range; ISAR, Identification of Seniors at Risk; MTS, Manchester Triage System; SD, Standard deviation. Triage categories are presented as per MTS.

*"Other" residential status refers to living arrangements not fitting existing categories (e.g., living with others, not family members).

**"Other" source of referral refers to referrals other than those listed.

***Index complaints are presented here from most to least common, independent of MTS category.

†"Other" complaints include abdominal pain; abscess and local infection; chest pain; collapsed adult; diarrhoea and vomiting; facial problems; gastrointestinal bleeds; head injury; headache; major trauma; shortness of breath; urinary problems; and wounds.

## Discussion

### Summary of findings

In this randomised controlled trial, early assessment and intervention by a dedicated HSCP team reduced the length of ED stay and incidence of hospital admission among older adults when compared to usual ED care. Patients in the intervention group reported higher satisfaction with the care received in the ED than in the control group, better function at 30-day and 6-month follow-up, better mobility (30 days), and better self-care (30 days and 6 months). No differences were observed at 30 days in terms of ED re-presentation, unscheduled hospital admission, or quality of life subdimensions of usual activities, discomfort, or anxiety. Considering objective outcomes at 6 months, patients in the intervention group had fewer unscheduled hospital admissions than those in the control group, but no significant differences were seen in terms of ED revisits or nursing home admissions.

**Table 3. Index visit outcomes.**

| Outcome | Control (n = 177) | Intervention (n = 176) | Adjusted OR (95% CI) p-value | Mean difference (95% CI) p-value | Mann–Whitney test |
|---|---|---|---|---|---|
| ED length of stay (hours), median (IQR) | 12.1 (6.18–22.14) | 6.43 (4.05–14.87) | | | Z = 4.82 $p < 0.001$ |
| Hospital admissions, n (%) | 99 (55.9) | 34 (19.3) | 0.18 (0.11, 0.31) $p < 0.001$ | | |
| Hospital length of stay (days), median (IQR) (n = 133) | 9 (5–24) | 9 (3–13) | | | Z = 0.99 $p = 0.32$ |
| Patient satisfaction, mean ± SD (n = 318) | 24.8 ± 3.74 | 25.8 ± 3.03 | | −1.01 (−1.76, −0.26) $p = 0.008$ | |

CI, confidence interval; IQR, interquartile range; OR, odds ratio; SD, standard deviation.

**Table 4. 30-day follow-up outcomes.**

| Outcome | Control | Intervention | Adjusted estimate (95% CI) |
|---|---|---|---|
| ED revisits, n (%) (n = 353) | 24 (13.6) | 33 (18.8) | OR = 1.42 (0.79, 2.55), p = 0.23 |
| Unscheduled hospital admissions, n (%) (n = 353) | 22 (12.4) | 21 (11.9) | OR = 0.96 (0.51, 1.84), p = 0.92 |
| Nursing home admissions, n (%) (n = 353) | 25 (14.1) | 17 (9.7) | OR = 0.76 (0.38, 1.51), p = 0.43 |
| Barthel index, mean ± SD (n = 323) | 15.56 ± 4.83 | 17.13 ± 3.52 | b = 0.97 (0.32, 1.61), p = 0.003, $R^2$ = 0.03 |
| EQ-5D-5L, mean ± SD (n = 329) | | | |
| Mobility | 2.27 ± 1.29 | 1.86 ± 1.11 | b = −0.29 (−0.52, −0.06), p = 0.02, $R^2$ = 0.03 |
| Self-care | 2.34 ± 1.48 | 1.89 ± 1.23 | b = −0.26 (−0.51, −0.02), p = 0.03, $R^2$ = 0.03 |
| Usual activities | 2.56 ± 1.36 | 2.22 ± 1.16 | b = −0.14 (−0.37, 0.08), p = 0.22, $R^2$ = 0.02 |
| Pain/discomfort | 1.95 ± 1.08 | 2.02 ± 1.06 | b = −0.01 (−0.22, 0.22), p = 0.97, $R^2$ = 0.01 |
| Anxiety/depression | 1.58 ± 0.92 | 1.46 ± 0.83 | b = −0.06 (−0.25, 0.11), p = 0.47, $R^2$ = 0.01 |

b, unstandardized coefficient adjusted for patient's age, baseline ISAR score, baseline function (for Barthel), and baseline quality of life (for quality of life); CI, confidence interval; ED, emergency department; EQ-5D-5L, EuroQoL 5-dimension 5-level; ISAR, Identification of Seniors at Risk; OR, odds ratio adjusted for patient's age and ISAR score at baseline; $R^2$, coefficient of determination; SD, standard deviation.

## Results in the context of existing evidence

This is, to the best of our knowledge, the first randomised controlled trial to evaluate the impact of a dedicated HSCP team on the care of older adults in the ED [16]. Evidence from existing studies supports the role of individual HSCPs in the ED in terms of reducing length of patient stay, avoiding unnecessary hospital admissions and improving the patient experience [16]. Furthermore, interdisciplinary care in the ED has been shown to enhance decision-making and contribute to timely and safe patient care, particularly for older adults [10–12]. Participants in our study who were allocated to the intervention group experienced significantly shorter ED stays owing to the timely assessment, intervention, and decision-making by an interdisciplinary HSCP team. Our finding of reduced hospital admissions in the intervention group resonate with a previous nonrandomised study that reported a decrease in hospital admissions among older adults assessed by an ED-based HSCP team [16], although we observed a much larger difference between intervention and control group.

**Table 5. 6-month follow-up outcomes.**

| Outcome | Control | Intervention | Adjusted estimate (95% CI) |
|---|---|---|---|
| ED revisits, n (%) (n = 346) | 74 (43.3) | 55 (31.4) | OR = 0.65 (0.42, 1.02), p = 0.06 |
| Unscheduled hospital admissions, n (%) (n = 346) | 57 (33.3) | 34 (19.4) | OR = 0.53 (0.32, 0.88), p = 0.02 |
| Nursing home admissions, n (%) (n = 304) | 11 (7.5) | 12 (7.6) | OR = 1.16 (0.48, 2.78), p = 0.74 |
| Barthel index, mean ± SD (n = 291) | 16.32 ± 4.28 | 17.29 ± 3.52 | b = 0.67 (0.02, 1.33), p = 0.04, $R^2$ = 0.02 |
| EQ-5D-5L*, mean ± SD (n = 297) | | | |
| Mobility | 2.04 ± 1.31 | 1.73 ± 1.11 | b = −0.21 (−0.45, 0.05), p = 0.11, $R^2$ = 0.02 |
| Self-care | 2.19 ± 1.43 | 1.72 ± 1.14 | b = −0.33 (−0.58, −0.08), p = 0.009, $R^2$ = 0.03 |
| Usual activities | 2.29 ± 1.31 | 2.02 ± 1.16 | b = −0.12 (−0.36, 0.12), p = 0.32, $R^2$ = 0.01 |
| Pain/discomfort | 1.73 ± 0.96 | 1.81 ± 0.99 | b = −0.01 (−0.22, 0.21), p = 0.97, $R^2$ = 0.001 |
| Anxiety/depression | 1.44 ± 0.72 | 1.31 ± 0.65 | b = −0.12 (−0.27, 0.04), p = 0.15, $R^2$ = 0.01 |

b, unstandardized coefficient adjusted for patient's age, baseline ISAR score, baseline function (for Barthel), and baseline quality of life (for quality of life); CI, confidence interval; ED, emergency department; EQ-5D-5L, EuroQoL 5-dimension 5-level; ISAR, Identification of Seniors at Risk; OR, odds ratio adjusted for patient's age and ISAR score at baseline; $R^2$, coefficient of determination; SD, standard deviation.

Our findings largely mirror those reported in a recent systematic review [31] that explored the impact of interventions aimed at improving clinical, patient experience, and healthcare utilisation for older adults presenting to the ED. Interventions across the 15 included studies (9 randomised controlled trials) were heterogeneous and comprised a variety of strategies (case management, discharge planning, and management/medication safety) delivered during the ED visit, following discharge and across the ED–primary care interface. While variation in the conduct and reporting of studies limited the strength of the conclusions drawn, when considering the interventions collectively, the authors reported a small benefit on functional status but no overall effect on ED return visit(s) or subsequent hospitalisation. Furthermore, interventions that bridged care occurring before and after ED discharge were associated with better outcomes. Our study did not include an intervention component that bridged the primary–secondary care interface and highlights the need to focus on interventions that integrate care before and after ED discharge.

Overall, we found that older adults across the intervention and control group reported high levels of satisfaction with care received in the ED, with those allocated to the intervention reporting significantly higher satisfaction with care than those in the control group. This finding aligns with those of previous studies [15,32,33], although the use of different assessment tools limits our ability to compare results. The follow-up differences observed in quality of life measures of mobility and self-care in contrast with those noted in Hughes and colleagues [31], although the studies included in their review used different measures of quality of life; this may point at the potential usefulness of assessing domain-specific aspects of quality of life.

## Strengths and limitations

In line with the MRC framework for the development and evaluation of complex interventions [20], our study employed a controlled and robust design to test the effectiveness of a low intensity model of care (short duration and limited number of patient contacts) in the ED. Also, the design trial was informed by a systematic review [16], key ED stakeholders' views [18], and a routine analysis of ED patient flow nationally to ensure the most optimal care pathways and inclusion criteria for the intervention. We assessed a comprehensive range of key outcomes to draw comparisons with existing literature and to provide conclusions on impact both at the patient and process level. Furthermore, our study was powered by an adequate sample size.

Our study is not without limitations. We focused on a specific cohort of "lower-urgency" patients with a range of index complaints that were amenable to HSCP intervention. This rationale was informed by our stakeholder input but, together with the fact that this is a single-centre trial, may limit the generalisability of the findings. We were unable to blind patients or ED staff members to allocation due to the nature of the intervention and the pragmatics of clinical practice in the ED, which may introduce a performance bias. Finally, our intervention was solely based in the ED and did not focus on care integration following discharge from the ED.

## Implications for practice and research

Older adults are frequent attendees at the ED and experience high rates of adverse outcomes following ED presentation. It is imperative that future research examines innovative models of care to meet the complex needs of this heterogeneous population. The findings of this study advance our knowledge on the clinical effectiveness of HSCP teams caring for older adults with a variety of complaints in the ED and support the viability of this model of care as a strategy to reduce ED length of stay and the risk of unnecessary hospital admissions for older individuals, who are at the highest risk of adverse outcome following an ED visit [8]. On the other

hand, the intervention did not reduce the incidence of ED re-presentations, suggesting the need for research to explore if integrated care packages involving community services to support older adults to live well in their own homes and communities are effective. Furthermore, a more in-depth exploration of the reasons for ED attendance among "lower-urgency" patients is warranted. It is possible that these patients may possess a lower threshold for attendance due to a myriad of factors including lower level of social and family support in the community and perceptions that the ED is reliable, accessible, and has the required expertise to deal with their presenting complaints [34,35].

## Conclusions

Identifying optimal intervention strategies to enhance the care of older adults in the ED is urgently needed in light of increasing ED attendances and associated negative impact on patient and process outcomes. In this randomised controlled trial, we found that early assessment and intervention by a dedicated HSCP team led to shorter stays in the ED, reduced risk of hospital admissions, and higher patient satisfaction at index visit. Our study thus supports the effectiveness of this model of care for key ED and patient outcomes.

## Supporting information

**S1 CONSORT Checklist. Consolidated Standards of Reporting Trials (CONSORT) checklist for the study.**
(DOC)

**S1 Intervention Characteristics. Characteristics of assessment and intervention by the health and social care team.**
(DOCX)

**S1 Sensitivity Analyses. Sensitivity analyses on outcomes ED length of stay and hospital admissions.**
(DOCX)

## Author Contributions

**Conceptualization:** Marica Cassarino, Katie Robinson, Íde O'Shaughnessy, Eimear Smalle, Stephen White, Rosie Quinn, Fiona Boland, Marie E. Ward, Rosa McNamara, Fiona Steed, Margaret O'Connor, Andrew O'Regan, Gerard McCarthy, Damien Ryan, Rose Galvin.

**Data curation:** Marica Cassarino, Collette Devlin.

**Formal analysis:** Marica Cassarino, Dominic Trépel, Collette Devlin, Fiona Boland, Rose Galvin.

**Funding acquisition:** Katie Robinson, Fiona Boland, Marie E. Ward, Rosa McNamara, Gerard McCarthy, Damien Ryan, Rose Galvin.

**Investigation:** Katie Robinson, Dominic Trépel, Íde O'Shaughnessy, Eimear Smalle, Stephen White, Rosie Quinn, Fiona Boland, Marie E. Ward, Rosa McNamara, Fiona Steed, Margaret O'Connor, Andrew O'Regan, Gerard McCarthy, Damien Ryan, Rose Galvin.

**Methodology:** Marica Cassarino, Dominic Trépel, Íde O'Shaughnessy, Eimear Smalle, Stephen White, Collette Devlin, Rosie Quinn, Fiona Boland, Marie E. Ward, Rosa McNamara, Fiona Steed, Margaret O'Connor, Andrew O'Regan, Gerard McCarthy, Damien Ryan, Rose Galvin.

**Project administration:** Marica Cassarino, Íde O'Shaughnessy, Eimear Smalle, Stephen White, Collette Devlin, Fiona Steed, Damien Ryan, Rose Galvin.

**Resources:** Íde O'Shaughnessy, Eimear Smalle, Stephen White, Rosie Quinn, Fiona Steed, Gerard McCarthy, Damien Ryan, Rose Galvin.

**Software:** Collette Devlin.

**Supervision:** Fiona Steed, Gerard McCarthy, Damien Ryan, Rose Galvin.

**Writing – original draft:** Marica Cassarino, Katie Robinson, Dominic Trépel, Rose Galvin.

**Writing – review & editing:** Marica Cassarino, Katie Robinson, Dominic Trépel, Íde O'Shaughnessy, Eimear Smalle, Stephen White, Collette Devlin, Rosie Quinn, Fiona Boland, Marie E. Ward, Rosa McNamara, Fiona Steed, Margaret O'Connor, Andrew O'Regan, Gerard McCarthy, Damien Ryan, Rose Galvin.

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
