## [Editor Report · Decision Letter 0]

4 Nov 2020

Dear Dr Cassarino, 

Thank you for submitting your manuscript entitled "Randomised controlled trial of the impact of a Health and Social Care Professional team in the emergency department on the quality, safety, and clinical effectiveness of care for older adults." for consideration by PLOS Medicine.

Your manuscript has now been evaluated by the PLOS Medicine editorial staff and I am writing to let you know that we would like to send your submission out for external assessment.

Kind regards,

Richard Turner, PhD

Senior editor, PLOS Medicine

rturner@plos.org

---

## [Decision Letter · Decision Letter 1]

19 Dec 2020

Dear Dr. Cassarino,

Thank you very much for submitting your manuscript "Randomised controlled trial of the impact of a Health and Social Care Professional team in the emergency department on the quality, safety, and clinical effectiveness of care for older adults." (PMEDICINE-D-20-05367R1) for consideration at PLOS Medicine. 

Your paper was evaluated by the editors and sent to independent reviewers, including a statistical reviewer. The reviews are appended at the bottom of this email and any accompanying reviewer attachments can be seen via the link below:

[LINK]

In light of these reviews, we will not be able to accept the manuscript for publication in the journal in its current form, but we would like to invite you to submit a revised version that addresses the reviewers' and editors' comments fully. You will appreciate that we cannot make a decision about publication until we have seen the revised manuscript and your response, and we expect to seek re-review by one or more of the reviewers. 

We hope to receive your revised manuscript by Jan 11 2021 11:59PM. Please email us (plosmedicine@plos.org) if you have any questions or concerns.

Please let me know if you have any questions. Otherwise, we look forward to receiving your revised manuscript in due course. 

Sincerely,

Richard Turner, PhD

rturner@plos.org

Please ensure that the data statement complies with PLOS' data policy (https://journals.plos.org/plosmedicine/s/data-availability), noting that study authors cannot be listed as points of contact for readers interested in inquiring about access to study data. 

Please adapt the title so that the study descriptor appears after a colon (i.e., "...: a randomized controlled trial"). 

Please restructure your abstract into a three-part structure preferred at PLOS Medicine. The final sentence of the "Methods and Findings" subsection should begin "Study limitations include ..." or similar and quote 2-3 of the study's main limitations. 

Please quote summary demographic characteristics for study participants in the abstract.

After the abstract, we will need to ask you to add a new and accessible "author summary" section in non-identical prose. You may find it helpful to consult one or two recent research papers in PLOS Medicine to get a sense of the preferred style. 

Please indicate in the methods section if consent was "informed". 

Please substitute "sex" for "gender" where appropriate. 

Throughout the text, please format reference call-outs as follows: "... poorer health outcomes [4,5].".

Please remove the information on funding, competing interests and data sharing from the end of the main text. In the event of publication, this information will appear in the article metadata via entries in the submission form.

Please ensure that all reference citations contain full access information (e.g., reference 4). 

Please correct the typo in reference 20.

Please add the full author names to reference 33. 

Please remove all publisher names (e.g., "Elsevier") and iterations of "[Internet]" from the reference list. 

Please adapt the attached CONSORT checklist so that individual items are referred to by section (e.g., "Methods") and paragraph number rather than by page or line numbers, as the latter generally change in the event of publication. Please refer to this in the text as "See S1_CONSORT_Checklist" or similar, and amend the attachment label to match. 

Comments from the reviewers:

*** Reviewer #1: 

This manuscript is well-written and easy to follow. I just have a few, relatively minor, comments listed below:

Looking at Table 2, patients randomised to the Control group appear to me as if being slightly less healthy that those randomised to the intervention group. For example, 43% vs 31% were identified as mildly frail or worse in the control and intervention groups respectively. I also note some differences in ISAR score and nutritional status. I would like the authors to consider describing these differences in the text as well as consider running sensitivity analyses that adjust for these baseline imbalances. I realise that a non-parametric test was used to assess differences in the primary outcome (length of ED stay); however, as a sensitivity analysis one could consider using a linear model which allows covariates.

Is it possible for the authors to add a table describing the intervention as delivered such as the types of activities performed and/or the time spent with the patient?

Please add the fact that the study was conducted in a single centre as one of the limitations (lack of generalisability).

Assuming that the same HSCP staff interacted with patient randomised to both arms, I would like to understand whether contamination (i.e. HSCP staff doing more under usual care than they would normally) occurred and what was put in place to prevent it.

A number of patients were excluded from the analysis of ED and hospital admissions at 30 days and 6 months due to death or lost-to-FU. Instead of excluding these patients, one could consider an alternative approach that take exposure/censoring into account. Possible methods include calculating rates (e.g. rates of admission) while adjusting for exposure or using a time-to-event analysis censored for death or loss-to-FU. One would need to consider how to handle death e.g. as a competing risk. I would encourage the authors to consider some of these options as a sensitivity analysis.

Please consider adding the pre-specified analysis plan in the appendix.

-Laurent Billot

*** Reviewer #2: 

please, find the enclosed file

*** Reviewer #3: 

General comments:

This is an interesting, important and timely study on improving post-ED outcomes for older adults seen in the ED utilizing a multidisciplinary HSCP team comprised of a social worker, occupational therapist and physiotherapist. The impressive reductions in ED length of stay and hospital admission rates, as well as improvements in 30-day function, mobility and self care generate enthusiasm for this potentially relatively uncomplicated intervention. However, one important aspect that limits the ability to apply and generalize the results to other settings is some description of what exactly the scope of the HSCP team was on patient care. Was it an assessment only with communication of findings to the medical team for them to determine treatment and follow-up care, an in-ED treatment intervention, an intervention including self-practice skills for home, a transition of care to an outpatient provider, or something else? The discussion suggests that transition of care did not occur, but more explicit description in the main methods and results would be important. If it was a combination, what percentage of cases fell into the various categories or multiple categories? How was a tailored approach determined if used? How long did the assessment and intervention take, and how many patients was the team actually able to intervene on in a day? Understanding the level of expertise and resources required will help inform how practical or translatable the content of the intervention is to other EDs.

Additional specific comments below for clarification of some of the sections:

Abstract: 

Participants: It would be helpful to include a couple words describing the inclusion criteria and/or whether this was a convenience sample, to give the reader a better sense from the abstract who the intervention was applied to. E.g. patients presenting between certain hours, and/or the overarching theme of the index complaints (e.g. illness or injury causing impairment of baseline functional status).

Introduction:

Page 6: The second sentence could be re-worded for clarity and grammar (e.g. incremental *increases* in ED attendance)

Page 7: First sentence of the second paragraph has multiple typographical and wording errors that make it confusing as to what the target outcome measures are. Please clarify to make it clear what the target outcomes are.

Page 8: Participants and recruitment: As in the abstract, it would be helpful to give a brief description of the theme of the inclusion criteria to better understand the target population (e.g. older adults whose illness or injury cause impairment of baseline functional status).

For the very urgent cases, would be helpful to specify what criteria were used after the work-up to deem them able to be invited (e.g. life-threatening causes ruled out, or likely to be discharged).

Page 10: Baseline data collection: For falls and hospitalizations, do you mean incidence (which is a proportion and usual population-based), or *number* of times for the individual? This seems like it should be "number"; if so, please re-word.

Page 11: Unclear what is meant by the term "bleeped". Please rephrase for clarity.

***

[LINK]

---

## [Decision Letter · Decision Letter 2]

8 May 2021

Dear Dr. Cassarino,

Thank you very much for submitting your revised manuscript "Impact of a Health and Social Care Professional team in the emergency department on the quality, safety, and clinical effectiveness of care for older adults: a randomised controlled trial" (PMEDICINE-D-20-05367R2) for consideration at PLOS Medicine. 

Your paper was discussed among the editors and sent to the reviewers, including our statistical reviewer. The reviews are appended at the bottom of this email and any accompanying reviewer attachments can be seen via the link below:

[LINK]

In light of these reviews, we will not be able to accept the manuscript for publication in the journal in its current form, but we would like to ask you to submit a further revised version that fully addresses the reviewers' and editors' comments. You will recognize that we cannot make a decision about publication until we have seen the revised manuscript and your response, and we plan to seek re-review by one or more of the reviewers. 

We hope to receive your revised manuscript by May 31 2021 11:59PM. Please email us (plosmedicine@plos.org) if you have any questions or concerns.

Please let me know if you have any questions, and we look forward to receiving your revised manuscript. 

Sincerely,

Richard Turner, PhD

rturner@plos.org

Please make that "data ... are" in your data statement (submission form). 

You may wish to note the element of assessment (by the team) in your title. 

Please substitute a colon for the semicolon in your title.

In the Methods section, please refer to figure 1 as "Participant flowchart" or similar rather than "CONSORT flowchart".

Please remove the information on ethics approval from the end of the main text (this belongs in the Methods section) and the information on transparency and dissemination.

Please use the abbreviation "PLoS ONE" in your reference list.

Comments from the reviewers:

*** Reviewer #1: 

Most of my comments have been adequately addressed; however, I have one remaining issue which relates to the adjustment for baseline imbalances.

The authors state that it was not necessary to adjust for differences in ISAR score, frailty or nutrition status given that tests comparing the two arms at baseline were not significant.

I believe that using significance tests to decide whether or not to adjust for baseline differences is potentially misleading due to the following reasons:

1) p-values measure the degree of consistency between the null hypothesis and the observed data. In the case of a randomised trial, we know that the null hypothesis is in fact true (i.e. that there is no true difference between the two randomised arms) and that any observed difference is due to chance.

2) One may not have the appropriate power to test for baseline differences

For these reasons, I believe it is important to decide whether or not to adjust, not based on tests of significance, but on observed differences. To quantify the differences, one option is to use standardised differences which do not depend on the sample size.

I would therefore strongly suggest adding the ISAR score, the frailty status and the nutrition status to the adjusted model already containing the triage category.

-Laurent Billot

*** Reviewer #2: 

Thank you for submitting the revised version of the manuscript. 

I thank the authors for taking into account all comments. The answers given are satisfactory. In particular, I appreciate the sensitivity analyses added, which reinforce the results. The description of the intervention characteristics is clear and detailed.

I have no more comments.

*** Reviewer #3: 

The revised manuscript is much clearer and addresses previous concerns.

***

[LINK]

---

## [Decision Letter · Decision Letter 3]

22 Jun 2021

Dear Dr. Cassarino,

Thank you very much for re-submitting your manuscript "Impact of assessment and intervention by a Health and Social Care Professional team in the emergency department on the quality, safety, and clinical effectiveness of care for older adults: a randomised controlled trial." (PMEDICINE-D-20-05367R3) for consideration at PLOS Medicine.

I have discussed the paper with editorial colleagues and it was also seen again by one reviewer. I am pleased to tell you that, once the remaining editorial and production issues are fully dealt with, we expect to be able to accept the paper for publication in the journal.

[LINK]

Please let me know if you have any questions, and we look forward to receiving the revised manuscript.   

Sincerely,

Richard Turner, PhD

rturner@plos.org

Requests from Editors:

Please quote aggregate demographic details for study participants in the abstract, i.e., age and sex from the rightmost column of table 2.

Where appropriate, e.g., in the abstract, please substitute "sex" for "gender".

In the author summary, please use the active voice (e.g., "We aimed ...) in at least one point. 

Comments from Reviewers:

*** Reviewer #1: 

The auhors have adequately addressed my last comment. I have no further comment.

-Laurent Billot

***

[LINK]

---

## [Editor Report · Decision Letter 4]

26 Jun 2021

Dear Dr Cassarino, 

On behalf of my colleagues, I am pleased to inform you that we have agreed to publish your manuscript "Impact of assessment and intervention by a Health and Social Care Professional team in the emergency department on the quality, safety, and clinical effectiveness of care for older adults: a randomised controlled trial." (PMEDICINE-D-20-05367R4) in PLOS Medicine.

Prior to final acceptance, please add a few words to your abstract to quote the study dates. 

PRESS

Sincerely, 

Richard Turner, PhD 

rturner@plos.org